# Long-Attention Weaving: Synthesizing Long-Context Data with High-Quality Short Data

## Abstract

The advent of long-context Large Language Models (LLMs) has been hindered by a critical bottleneck: the scarcity of high-quality training data. Standard data synthesis methods, which typically concatenate short documents, often fail to create the challenging, long-range dependencies essential for robust learning. In this work, we introduce **Long-Attention Weaving (LAW)**, a novel framework that leverages a model's own self-attention mechanism to synthesize a superior long-context training curriculum. LAW operates in two stages: first, it employs a multi-scale attention-based score to identify short documents that are inherently rich in long-range dependencies. Second, it utilizes a novel *interleaving* strategy to weave these selected documents into complex sequences, compelling the model to establish non-trivial, long-distance relationships. We demonstrate that continually pre-training LLaMA-2 7B on data synthesized by LAW extends its effective context window to 64k and significantly outperforms strong baselines on a suite of long-context benchmarks, LongBench. Our findings highlight the efficacy of attention-guided data engineering for unlocking the full potential of long-context LLMs. All code and data are available at https://anonymous.4open.science/r/LAW-B056.

## 1 Introduction

The capacity to process extensive contexts is a critical frontier in the advancement of Large Language Models (LLMs), underpinning their application in complex, real-world domains such as comprehending entire codebases, summarizing lengthy legal documents, or engaging in multi-turn dialogues over long histories (Vaswani et al., 2017). While proprietary models like GPT-4 and Claude 3 have demonstrated remarkable long-context capabilities (OpenAI, 2023; Anthropic, 2024), the open-source community's efforts to replicate this success are often hampered by a fundamental bottleneck: the scarcity of high-quality, naturally occurring long-text data (Liu et al., 2023; Bai et al., 2023).

To circumvent this data shortage, a common practice is to synthesize long training samples by concatenating shorter documents from large corpora (Raffel et al., 2020; Gao et al., 2020), a strategy employed during the pre-training of many foundational models (Brown et al., 2020; Touvron et al., 2023). However, this approach often creates an "illusion" of long context—sequences that are long in token count but lack the deep, interwoven dependencies that characterize genuine long-form text (Bai et al., 2023). Models trained on such data can often achieve low perplexity without developing the sophisticated, long-range reasoning skills they are intended to acquire, as the necessary information is frequently found within local, more easily accessible context windows (Liu et al., 2023; Xiao et al., 2023).

This challenge has spurred two main lines of research, both of which have significant limitations. The first, which we term **brute-force synthesis**, attempts to create more challenging data by interleaving multiple short documents, as seen in methods like LongSkywork (Zhao et al., 2024). While an improvement over simple concatenation, this strategy treats all source documents as equally valuable, inevitably diluting the training corpus with material that lacks strong internal dependencies and thus offers a weak learning signal (Swayamdipta et al., 2020).

The second line of work focuses on **misaligned filtering**. These methods aim to select higher-quality data but rely on flawed proxies. Linguistic-based approaches like ProLong (Cheng et al., 2024) use

metrics such as perplexity to filter data, a technique which is widely used in web-scale data cleaning (Wenzek et al., 2019; Raffel et al., 2020), yet one which is computationally intensive and poorly aligned with the token-level attention mechanisms that govern LLM processing. More model-aware methods, such as LongAttn (Wu et al., 2025), cleverly use self-attention scores to find dependency-rich texts, drawing on the idea that models can report on their own internal states (Kadavath et al., 2022). However, their framework is designed to filter a corpus of *already-long* documents, not to guide the synthesis of new long documents from a pool of short ones, making it ill-suited for the data scarcity problem.

In this paper, we introduce **Long-Attention Weaving (LAW)**, a new framework that moves beyond these limitations by directly using an LLM's internal self-attention scores to both select high-quality source material and synthesize a challenging long-context curriculum. Our core insight is that the most effective data forces the model to learn relationships that it currently struggles with, a signal best captured by its own attention mechanism. LAW implements this via a two-stage process:

- We first design a multi-scale, attention-based scoring metric to efficiently identify short documents that are rich in internal, long-range dependencies, ensuring that every component of our synthetic data is of high quality.
- We then employ a novel *interleaving* synthesis strategy that weaves these selected documents into complex sequences, constructing complex sequences that explicitly require the model to connect semantically related but distant segments of text.

We demonstrate through extensive experiments that continually pre-training LLaMA-2 7B on data synthesized by LAW significantly enhances its long-context capabilities on a diverse array of downstream benchmarks. Our contributions are not just a new method, but a new perspective: treating long-context data synthesis as a data engineering problem, guided by the model's own perception of contextual dependency.

## 2 RELATED WORK

The pursuit of longer context windows in LLMs has advanced along two primary axes: architectural innovations and data-centric strategies. Our work falls into the latter, focusing on the generation of high-quality training data, a component we argue is critical for any architecture to realize its full potential.

**Architectural Innovations** A significant body of research has focused on modifying model architectures to handle longer sequences. Key efforts include developing more efficient attention mechanisms to mitigate the quadratic complexity of standard self-attention (Zhuang et al., 2023). These "X-formers" encompass a wide range of techniques, such as sparse attention patterns (Beltagy et al., 2020; Zaheer et al., 2020), hardware-aware optimizations like FlashAttention (Dao et al., 2022), and alternative architectures like state-space models that offer linear-time complexity (Gu & Dao, 2023). Another critical area is the adaptation of positional encodings to extrapolate beyond their original training length. Techniques such as Positional Interpolation (PI) (Chen et al., 2023) and YaRN (Peng et al., 2023) have become standard practices for extending the context window of existing models like LLaMA. However, these architectural modifications only provide the *capacity* for longer contexts; the model must still learn to *utilize* this capacity through exposure to appropriate data, a challenge that remains significant. Our work is orthogonal and complementary to these efforts, providing the high-quality data needed to make such architectural extensions effective.

**Data-Centric Strategies** The performance of long-context models is intrinsically linked to the data they are trained on (Touvron et al., 2023). Given the scarcity of naturally long, high-quality documents, research in this area has broadly bifurcated into data synthesis and data selection.

**Data Synthesis via Document Combination.** A common and scalable strategy for creating long-text data is to combine shorter documents from large-scale corpora like The Pile (Gao et al., 2020), C4 (Raffel et al., 2020), and BookCorpus (Zhu et al., 2015). The simplest form is sequential concatenation, a method used in the pre-training of many foundational models (Brown et al., 2020; Touvron et al., 2023). A more advanced technique, employed by LongSkywork (Zhao et al., 2024), uses a chunk-interleaving strategy that weaves segments from multiple documents together. This forces

the model to track information from different sources simultaneously, creating a more challenging training task than simple concatenation. However, by treating all source documents as equally valuable, these synthesis-only approaches risk diluting the training data with samples that lack strong internal dependencies, potentially leading to inefficient or ineffective learning.

**Data Selection via Quality Scoring.** To address the issue of data quality, other methods focus on selecting the most valuable data from a corpus, a practice philosophically rooted in ideas like dataset cartography (Swayamdipta et al., 2020) and influence functions (Koh & Liang, 2017). Early approaches like ProLong (Cheng et al., 2024) used linguistic metrics like perplexity to score and select coherent document sequences, a technique also common in large-scale data cleaning pipelines (Wenzek et al., 2019). While intuitive, such metrics are often computationally expensive and are not directly aligned with the internal mechanisms of the transformer architecture. A more direct and aligned approach was pioneered by LongAttn (Wu et al., 2025), which leverages the model's own self-attention scores to identify existing long documents rich in long-range dependencies, building on the insight that models can possess self-knowledge about their own internal states (Kadavath et al., 2022). This was a key insight, but its methodology is designed as a *filter* for an existing corpus of long texts. It does not address the core problem of how to *synthesize* new long documents when a large, high-quality long-text corpus is not readily available.

**Positioning of Long-Attention Weaving** Our work, Long-Attention Weaving (LAW), synergizes and advances these two paradigms. We adopt the synthesis-centric approach of methods like LongSkywork, but address its core limitation by introducing a critical data selection stage. Our selection mechanism is inspired by the model-centric philosophy of LongAttn, but we make two crucial adaptations: (1) we apply it to score *short documents* to assess their suitability as building blocks for synthesis, and (2) we enhance the scoring with a multi-scale analysis and a robust rank-aggregation scheme. By first selecting for dependency-rich short documents and then weaving them together using a challenging interleaving strategy, LAW creates a more potent and efficient training curriculum specifically designed to foster long-range reasoning.

## 3 METHODOLOGY

The core of our framework is a three-stage pipeline designed to generate a high-quality, long-context training curriculum from a large corpus of short documents. The stages are: (1) multi-range context dependency scoring of short documents, (2) multi-scale ranking aggregation for robust selection, and (3) long-context synthesis via document interleaving.

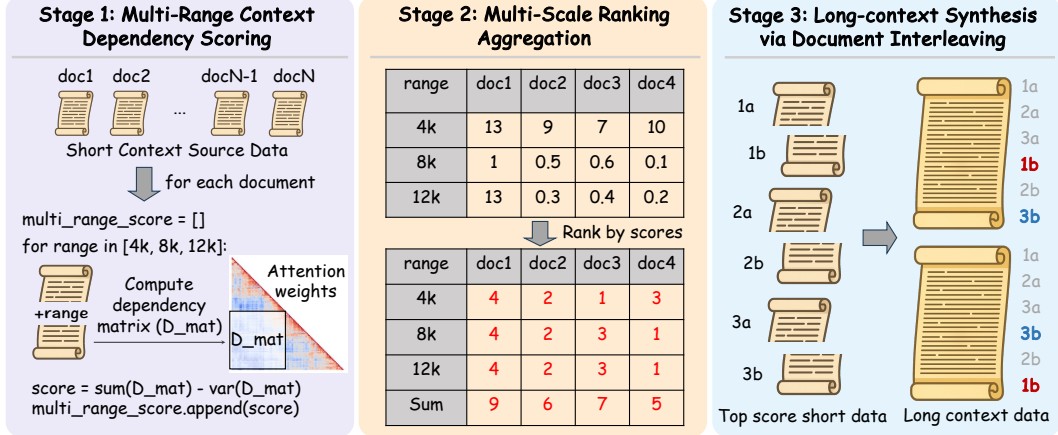

Figure 1: The overall framework of Long-Attention Weaving (LAW) consists of three stages. Stage 1: Short-context documents are scored for long-range dependency richness at multiple ranges (e.g., 4k, 8k, 12k) using the first-layer self-attention matrix to compute multi-range scores. Stage 2: Scores from each range are converted to ranks and aggregated via summation to yield a final score for robust document selection. Stage 3: Top-scoring short documents are bisected and interleaved using ordered and reverse-ordered strategies to synthesize challenging long-context training data.

### 3.1 STAGE 1: MULTI-RANGE CONTEXT DEPENDENCY SCORING

Our process begins with a large-scale corpus of short documents, primarily sourced from open-source code repositories and ArXiv papers due to their rich semantic content. Raw documents are segmented into fixed-length chunks (e.g., 4k tokens) using a sliding-window strategy designed to maximize data utilization while preserving informational integrity. For a document of $n$ tokens and a target length $L$, we extract chunks from the beginning, end, and middle, ensuring that no content is wasted and that each chunk represents a contiguous block of text. This forms the initial pool of short-context documents, $\mathcal{D}$.

To identify the most valuable documents for synthesis, we introduce a scoring mechanism that quantifies long-range dependencies from the model's own perspective. Our approach extends the core idea of LongAttn (Wu et al., 2025) by adapting it to score short documents and incorporating a multi-range analysis for robustness.

The process begins by computing a Long-Range Dependency Score (LDS) for each document $s \in \mathcal{D}$ at multiple distance thresholds $k \in K$. The score is derived from the self-attention matrix $\mathbf{M}$ of a pre-trained LLM, extracted specifically from its first layer. This choice is deliberate: the first layer is computationally efficient and its attention patterns are less influenced by task-specific heads or the "attention sink" phenomenon, providing a purer signal of fundamental token-level relationships (Xiao et al., 2023). The score is defined as:

$$\text{LDS}(s, k) = \text{Mean}(\mathbf{M}_{i,j|j-i>k}) - \alpha \cdot \text{Var}(\mathbf{M}_{i,j|j-i>k}) \tag{1}$$

where the first term is the average attention strength beyond distance $k$, and the second term penalizes non-uniform dependency distributions with a hyperparameter $\alpha$.

### 3.2 STAGE 2: MULTI-SCALE RANKING AGGREGATION

A critical challenge in utilizing our multi-range dependency scores is that the raw LDS values are not directly comparable across different scales $k$. The distribution and magnitude of attention scores can vary significantly between short-range and long-range dependencies; a naive summation would be susceptible to biases, allowing a single scale to dominate the final score. To mitigate this issue of incommensurability, we adopt a robust, non-parametric approach by converting the raw scores into ranks. For each distance scale $k \in K$, we rank all documents $s \in \mathcal{D}$ according to their $\text{LDS}(s, k)$ values, yielding a set of rank lists, $R(s, k)$.

These individual rank lists are then aggregated into a single, unified score using a method inspired by the Borda Count(Emerson, 2013)consensus system. In this formulation, each scale $k$ acts as a "voter," and the final score for a document reflects its overall standing across all voters. This is achieved by summing the ranks a document received across all scales:

$$\text{Score}_{\text{final}}(s) = \sum_{k \in K} R(s, k) \tag{2}$$

This rank-aggregation strategy is inherently robust, as it prioritizes documents that achieve a strong consensus of high performance across the entire spectrum of dependency ranges. It rewards documents that are consistently ranked favorably, rather than those that may have an exceptionally high score at one scale but perform poorly at others.

Finally, we perform a principled filtering step by selecting the top $p$-th percentile (e.g., top 50%) of documents based on their $\text{Score}_{\text{final}}(s)$. This process yields the high-quality subset $\mathcal{D}^*$, which serves as the source material for our subsequent synthesis stage.

### 3.3 STAGE 3: LONG-CONTEXT SYNTHESIS VIA DOCUMENT INTERLEAVING

Using the high-quality document set $\mathcal{D}^*$, we synthesize long-context samples designed to be challenging for the model. The process begins by sampling a batch of $N$ documents (e.g., $N = 8$). Each document $D_i$ is bisected into two halves, $P_{i,1}$ and $P_{i,2}$.

We employ two interleaving strategies. The first, **Ordered-Interleaving**, concatenates all first-half parts followed by all second-half parts in their original order, creating a structure that requires the

model to connect related but now distant halves of the same documents:

$$D_{\text{synth\_ordered}} = [P_{1,1} \circ \cdots \circ P_{N,1} \circ P_{1,2} \circ \cdots \circ P_{N,2}] \tag{3}$$

Our primary strategy, **Reverse-Ordered Interleaving**, introduces a greater challenge by reversing the order of the second-half parts. This breaks simple positional heuristics the model might learn and forces a more robust, content-based understanding of semantic links to solve the complex "semantic binding" task of reconnecting the document halves:

$$D_{\text{synth\_reversed}} = [P_{1,1} \circ \cdots \circ P_{N,1} \circ P_{N,2} \circ \cdots \circ P_{1,2}] \tag{4}$$

where $\circ$ denotes concatenation. The final training set is composed of synthetic documents generated using a mix of these strategies, creating a diverse and challenging curriculum for long-context learning.

The final stage of our framework utilizes the synthesized long-context corpus for the continual pre-training of a base LLM. The primary training objective is to extend its effective context window. The central logic of our data generation process is formalized in Algorithm 1.

## 4    EXPERIMENTS

To validate the efficacy of our proposed framework, Long-Attention Weaving (LAW), we conduct a comprehensive suite of experiments. We first detail the experimental setup, then present the main results comparing LAW against strong baselines, and conclude with in-depth ablation studies and qualitative analyses to deconstruct the sources of its performance gains.

### 4.1    EXPERIMENTAL SETUP

**Training Details**    All models were initialized from the LLaMA-2 7B checkpoint and continually pre-trained on a corpus of 1 billion (1B) tokens. This corpus was synthesized by sampling from a curated dataset of ArXiv papers and source code at a 1:1 ratio. We extended the model's context window from its native 4k to 64k tokens. Following established best practices for long-context adaptation (Lu et al., 2024), we employed a learning rate of $2 \times 10^{-5}$ with a linear warmup schedule and no weight decay. All experiments were conducted on a cluster of 8 NVIDIA H800 GPUs.

**Baselines**    We benchmark LAW against three strong baselines designed to isolate the contributions of our framework's core components:

- **Base Model:** The original LLaMA-2 7B checkpoint, without any long-context continual pre-training, serving as a reference for pre-adaptation performance.
- **Random Concatenation:** A naive baseline that bypasses our attention-based filtering stage. Short documents are randomly sampled and concatenated to form 64k-token sequences, representing a common but simplistic approach to long-context data construction.
- **Ordered-Interleaving w/o Filter:** This baseline ablates our filtering mechanism by applying the ordered-interleaving strategy to randomly selected documents. It serves to isolate the performance gains attributable specifically to our attention-guided document selection.

**Evaluation Tasks**    We employ a diverse set of intrinsic and extrinsic evaluation tasks to provide a holistic assessment of model capabilities.

- **Intrinsic Evaluation:** We measure perplexity (PPL) on the PG19 and Proof-pile datasets to assess fundamental language modeling quality across various context lengths.
- **Extrinsic Evaluation:** We report performance on LongBench (Bai et al., 2023), a standard multi-task benchmark for long-context understanding. Additionally, we assess in-context learning ability on the Trec News dataset using a "ManyShots" learning paradigm.

### 4.2    MAIN RESULTS

**Language Modeling Perplexity**    Table 1 presents the perplexity scores on the PG19 and Proof-pile datasets. At the target context length of 64k, LAW achieves superior language modeling performance, confirming its effectiveness in modeling long-range dependencies.

A noteworthy observation is LAW's slightly elevated perplexity at shorter context lengths (e.g., 2k-8k) compared to the baselines. We posit that this is not a deficiency but rather an expected artifact and a positive indicator of our method's success. The data synthesized by LAW is intentionally structured to be complex and non-locally redundant, compelling the model to resolve dependencies that span beyond short evaluation windows. Consequently, when evaluated on these shorter windows, the model's attempts to predict tokens based on unavailable long-range context may result in a marginal PPL increase. In contrast, baseline models trained on locally coherent but globally simplistic concatenated data excel at local prediction but fail to develop the mechanisms for genuine long-range reasoning, as evidenced by their degraded performance at the full 64k context length.

Table 1: Perplexity (PPL) on PG19 and Proof-pile datasets at various context lengths. All models are trained with a 64k context window. Lower PPL indicates better performance. LAW (Ours) achieves the best performance at the longest context lengths, validating its long-range modeling capabilities.

| Dataset | Model | 2k | 4k | 8k | 16k | 32k | 64k |
|---------|-------|-----|-----|-----|------|------|------|
| PG19 | Random Concatenation | **7.11** | **6.73** | **6.50** | 6.36 | 6.25 | 6.20 |
| | Ordered-Interleaving w/o Filter | 7.25 | 6.84 | 6.57 | 6.40 | 6.27 | 6.19 |
| | **LAW (Ours)** | 7.41 | 7.29 | 6.62 | **6.45** | **6.31** | **6.18** |
| Proof-pile | Random Concatenation | **3.28** | **3.00** | 2.82 | 2.68 | 2.59 | 2.53 |
| | Ordered-Interleaving w/o Filter | 3.29 | 3.01 | 2.83 | 2.67 | 2.57 | 2.50 |
| | **LAW (Ours)** | 4.63 | 4.67 | **2.83** | **2.67** | **2.57** | **2.49** |

**Downstream Task Performance** As shown in Table 2, LAW consistently and significantly outperforms all baselines on the LongBench benchmark average score. This result underscores the tangible benefits of our attention-guided data synthesis strategy for downstream applications. The substantial gains over both Random Concatenation and Ordered-Interleaving without Filter highlight that both components of our framework—the principled selection of documents with high dependency potential and the challenging interleaving synthesis strategy—are critical for achieving state-of-the-art performance.

Crucially, the superior performance is not merely in information retrieval but extends to tasks requiring deeper reasoning, such as Question Answering and Summarization. We attribute this to the nature of the training signal provided by LAW. The interleaving process forces the model to disentangle, track, and integrate information from multiple, non-contiguous sources within a single context, thereby directly training the cognitive primitives necessary for complex, multi-hop reasoning.

### 4.3 ABLATION STUDIES AND ANALYSIS

**Impact of Framework Components** To deconstruct the contributions of LAW's key components, we conducted targeted ablation studies, with results presented in Table 3.

- **Ablation on Document Filtering:** Isolating the effect of our attention-based filtering by comparing the full LAW model to the "Ordered-Interleaving w/o Filter" baseline reveals its criticality. The performance degradation (from 31.26 to 30.60 on LongBench) confirms that our scoring mechanism is highly effective at identifying documents that provide a rich training signal for learning long-range dependencies.

- **Ablation on Synthesis Strategy:** The performance gap between LAW and Random Concatenation (31.26 vs. 31.05) underscores the efficacy of the interleaving synthesis method. To further probe this, we trained a variant, 'LAW w/o Reverse-Order', which omits the reverse-ordered interleaving component. As shown in Table 3, this variant underperforms the full method, validating that the increased complexity introduced by the bidirectional interleaving strategy is a key contributor to the model's final performance.

**Sensitivity and Robustness Analysis** We analyze the robustness of our method with respect to key design choices.

Table 2: Main results on the LongBench benchmark, averaged across task categories. LAW (Ours) demonstrates substantial improvements over baselines, highlighting the effectiveness of our data synthesis framework in fostering advanced reasoning capabilities.

| Category | Task | Base Model | Rand. Concat. | Ord. Interleaving w/o Filter | LAW (Ours) |
|---|---|---|---|---|---|
| Question Answering | NQA | 20.46 | 22.73 | 22.31 | 22.43 |
| | QAP | 28.08 | 28.56 | 29.22 | 28.53 |
| | MQA | 37.72 | 36.44 | 39.27 | 40.94 |
| | HQA | 42.03 | 41.61 | 39.56 | 40.58 |
| | WQA | 30.11 | 34.25 | 31.60 | 31.92 |
| | TQA | 85.31 | 87.27 | 86.92 | 86.76 |
| Summarization | MSQ | 14.40 | 14.75 | 18.27 | 17.74 |
| | QSM | 20.93 | 20.02 | 20.37 | 20.47 |
| | MWS | 14.47 | 15.62 | 13.12 | 17.13 |
| | SMS | 41.20 | 40.74 | 41.90 | 41.75 |
| Code | PSC | 2.68 | 1.18 | 1.45 | 2.55 |
| | PSR | 8.75 | 6.05 | 5.91 | 5.55 |
| | LCC | 23.10 | 32.98 | 16.67 | 18.07 |
| | REP | 26.54 | 29.54 | 28.31 | 29.79 |
| Other | GR | 21.66 | 15.61 | 24.73 | 25.96 |
| | TRE | 70.50 | 69.50 | 70.00 | 70.00 |
| **Average** | | **30.50** | **31.05** | **30.60** | **31.26** |

Table 3: Ablation and data scaling results on LongBench (average score). These results demonstrate the positive impact of scaling training data and confirm the benefit of our full reverse-ordered interleaving strategy.

| Model | LAW (Ours, 1B) | LAW (2B Tokens) | LAW (w/o Reverse-Order) |
|---|---|---|---|
| Avg. Score | 31.26 | 32.14 | 31.06 |

- **Impact of Training Data Scale:** By training a model on a 2B-token corpus synthesized by LAW, we observe a consistent performance improvement (from 31.26 to 32.14 on Long-Bench), demonstrating the scalability and positive data-scaling properties of our framework.

- **Generalization Across Context Lengths:** The perplexity results in Table 1 show a graceful degradation as the evaluation context shortens (e.g., 6.18 at 64k vs. 6.31 at 32k on PG19). This suggests the learned long-context capabilities are robust and not pathologically tied to the maximum training length.

- **Orthogonality to Position Extrapolation Method:** To ensure our data-centric improvements are not conflated with architectural choices, we trained models using LAW-synthesized data with two alternative RoPE extrapolation techniques: Positional Interpolation (PI) and YaRN. In all configurations, training on LAW data yielded significant improvements over baselines. This confirms that our synthesis method provides benefits that are largely orthogonal to and complementary with advances in positional encoding strategies.

### 4.4 QUALITATIVE ANALYSIS OF ATTENTION MECHANISMS

To provide qualitative evidence for how LAW shapes the model's reasoning, we visualize its attention patterns. Figure 2 compares heatmaps from documents with high and low dependency scores, as determined by our filtering mechanism.

As illustrated in Figure 2a, documents with high dependency scores elicit dense, non-local attention patterns. The strong off-diagonal signals, particularly in the lower-left quadrant, indicate that tokens late in the sequence are actively attending to foundational concepts introduced much earlier. This is a hallmark of sophisticated, long-range reasoning and validates that our scoring metric successfully identifies texts that demand such behavior.

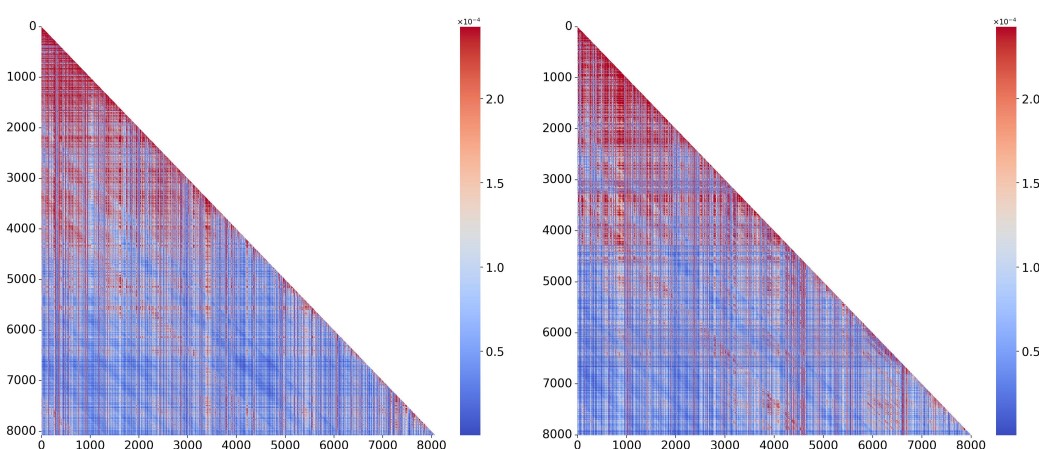

(a) High dependency score documents exhibit strong off-diagonal attention, indicating a focus on long-range dependencies.

(b) Low dependency score documents show attention concentrated on the diagonal, signifying a reliance on local context.

Figure 2: A comparison of attention patterns on the LAW dataset. The model dynamically adjusts its focus from long-range (a) to short-range (b) dependencies based on the document's structural complexity.

Conversely, for documents with low dependency scores (Figure 2b), the attention pattern converges to a band-diagonal structure. This signifies a reliance on local context, where tokens primarily attend to their immediate neighbors, reflecting a more sequential and less integrative processing mode.

These visualizations provide compelling evidence that our training framework does not merely extend the context window but instills a **dynamic, input-dependent attentional strategy**. The model learns to allocate its cognitive resources efficiently, shifting from a global, integrative focus for complex documents to a local, sequential focus for simpler ones. This adaptive capability is a direct result of being trained on a curriculum curated and structured by LAW.

### 4.5    ABLATION STUDY ON DATA SYNTHESIS STRATEGY

To further dissect the contribution of our proposed data synthesis method, we conduct an ablation study focusing on the 'reverse' operation. We visualize the attention patterns of models trained on data synthesized with our full methodology versus a variant where the 'reverse' step is omitted. This comparison, presented in Figure 3, serves to highlight the impact of this specific component on the model's ability to capture long-range dependencies.

Figure 3a displays the attention heatmap from a model trained on data synthesized using our complete method. The heatmap is characterized by a pronounced and vibrant signal in the lower-left quadrant. This indicates that tokens appearing later in the sequence (represented by the y-axis) are assigning high attention weights to tokens from the very beginning of the sequence (x-axis). Such a pattern is a definitive indicator of robust long-range dependency modeling, as it shows the model's capacity to maintain and access context over extended distances.

In contrast, Figure 3b illustrates the attention pattern from a model trained on data synthesized without the 'reverse' operation. While off-diagonal attention is still present, the intensity of the signal in the lower-left quadrant is visibly diminished compared to Figure 3a. The reduced attention scores in this critical region suggest a comparative weakening in the model's ability to form connections between distant tokens. The 'reverse' operation, by forcing the model to predict the beginning of a sequence from its end, explicitly trains the model to integrate information across the entire context length, thereby strengthening these long-range dependencies.

This qualitative comparison underscores the efficacy of our full data synthesis approach. The heightened attention scores in the lower-left quadrant of Figure 3a are not merely an artifact but a direct visualization of the model's enhanced capacity for long-range reasoning—a capacity specifically

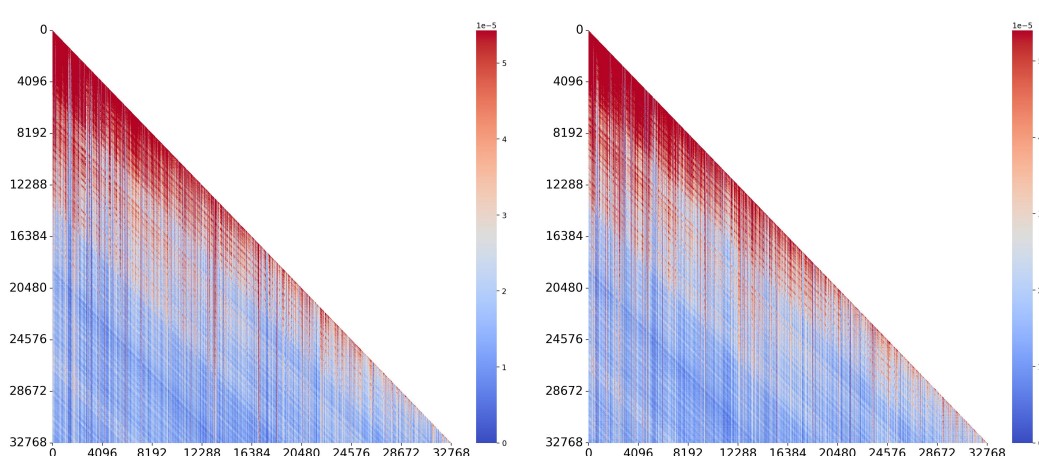

(a) Full synthesis method. The pronounced attention in the lower-left quadrant demonstrates strong long-range dependencies.

(b) Synthesis method without 'reverse' operation. The attenuated signal in the lower-left indicates a reduced focus on long-range dependencies.

Figure 3: Ablation study of attention patterns. A comparison of our full data synthesis method (a) with a variant lacking the 'reverse' operation (b). The full method results in markedly stronger long-range attention.

cultivated by the inclusion of the 'reverse' operation in our data synthesis pipeline. This provides strong evidence that our complete method is superior for instilling the desired long-range dependency capabilities in the model.

## 5  CONCLUSION

In this work, we introduced Long-Attention Weaving (LAW), a new framework for synthesizing a high-quality, long-context training curriculum. Our core contribution is a model-centric approach to data engineering: we leverage an LLM's own self-attention mechanism to both identify short documents rich in long-range dependencies and weave them into complex synthetic sequences that promote robust learning. Through extensive experiments, we demonstrated that training a LLaMA-2 7B model on data generated by LAW leads to significant improvements in perplexity and downstream performance on a wide range of long-context benchmarks. The success of LAW underscores a critical principle: for long-context learning, the structure and quality of data are as important as architectural innovations. Our findings suggest that future progress in this domain will increasingly rely on sophisticated, model-aware data synthesis strategies. This work represents a step in that direction, opening up promising avenues for research into attention-guided curriculum learning, the interplay between synthetic and natural data distributions, and the automated creation of challenging training regimes that push the boundaries of what LLMs can achieve.

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

## A    APPENDIX

### A.1    LONG-ATTENTION WEAVING (LAW) DATA SYNTHESIS PROCESS

---

**Algorithm 1** Long-Attention Weaving (LAW) Data Synthesis Process

---

1: **Input:** Short document corpus $\mathcal{D}$, scoring model $\mathcal{M}$, distance scales $K = \{k_1, \ldots, k_m\}$, selection percentile $p$, synthesis batch size $N$.
2: **Output:** A batch of synthetic long documents $\mathcal{B}_{\text{synth}}$.
3:
4:                                               ▷ *Stage 1: Multi-Range Context Dependency Scoring*
5: Initialize score list $\mathcal{S}_k$ for each $k \in K$.
6: **for** each document $s \in \mathcal{D}$ **do**
7:      Extract first-layer attention matrix $\mathbf{M}$ from $\mathcal{M}(s)$.
8:      **for** each scale $k \in K$ **do**
9:          Compute $\text{LDS}(s, k)$ from $\mathbf{M}$ using Equation 1.
10:          Append score to $\mathcal{S}_k$.
11:      **end for**
12: **end for**
13:
14:                                               ▷ *Stage 2: Multi-Scale Ranking Aggregation*
15: **for** each scale $k \in K$ **do**
16:      Convert scores $\mathcal{S}_k$ to ranks $R(s, k)$ for all $s \in \mathcal{D}$.
17: **end for**
18: **for** each document $s \in \mathcal{D}$ **do**
19:      Calculate $\text{Score}_{\text{final}}(s)$ using Equation 2.
20: **end for**
21: Create high-quality set $\mathcal{D}^*$ by selecting top $p\%$ of documents from $\mathcal{D}$ based on $\text{Score}_{\text{final}}$.
22:
23:                                               ▷ *Stage 3: Long-Context Synthesis via Document Interleaving*
24: Initialize batch $\mathcal{B}_{\text{synth}} = []$.
25: Sample $N$ documents $\{D_1, \ldots, D_N\}$ from $\mathcal{D}^*$.
26: Bisect each document $D_i$ into parts $P_{i,1}$ and $P_{i,2}$.
27: Construct $D_{\text{synth\_ordered}}$ and $D_{\text{synth\_reversed}}$ (Eq. 3 and 4).
28: Add synthesized documents to $\mathcal{B}_{\text{synth}}$.
29: **return** $\mathcal{B}_{\text{synth}}$

---

### A.2    LIMITATIONS

While Long-Attention Weaving demonstrates significant efficacy, we acknowledge several limitations that offer avenues for future research. First, our framework relies on a pre-trained model's self-attention scores as a proxy for semantic dependency. While we argue this is more aligned than external linguistic metrics, these attention scores can be noisy and may not perfectly capture all forms of long-range relationships, particularly abstract or inferential ones. The quality of the synthesized data is therefore inherently tied to the capabilities of the initial scoring model.

Second, the data synthesis process, particularly the interleaving strategy, creates a distribution of text that is structurally different from naturally occurring long documents. While we have shown this to be a powerful training signal, it may introduce a subtle domain mismatch, potentially leading the model to develop biases or heuristics optimized for this synthetic structure. An important future direction is to investigate methods for gradually annealing the training curriculum from synthetic, interleaved data towards more natural long-form text.

Finally, our multi-scale scoring approach, while more robust than single-scale methods, introduces hyperparameters related to the choice of distance thresholds ($K$) and the selection percentile ($p$). Although our experiments show strong performance with a standard configuration, a more systematic exploration of these hyperparameters could yield further improvements and a better understanding of their impact on different data modalities.

### A.3 USAGE OF LLMS FOR WRITING

We use LLMs to aid or polish writing the whole paper.

### A.4 RESPONSE EXAMPLES OF THE TRAINED MODEL ON LONGBENCH

In this subsection, we present illustrative response examples from our trained model on the LAW dataset. These cases demonstrate the model's proficiency in managing long-range dependencies, synthesizing coherent narratives from interleaved document segments, and adapting to complex semantic structures. Each example highlights different aspects of the model's long-context capabilities, such as multi-hop reasoning, information retrieval across distant contexts, and robust handling of reversed interleaving patterns.

---

**Example 1**

**NarrativeQA**: You are given a story, which can be either a novel or a movie script, and a question. Answer the question as concisely as you can, using a single phrase if possible. Do not provide any explanation.

Story: {Here is the story.}

Now, answer the question based on the story as concisely as you can, using a single phrase if possible. Do not provide any explanation.

Question: {input}

**Ours**: A security guard.

**Ground truth**: A security guard.

**Prompt length**: 29630

---

Figure 4: Model response in the first case from the LongBench dataset, showcasing effective resolution of long-range semantic dependencies in an interleaved context.

**Example 2**

**NarrativeQA**: You are given a story, which can be either a novel or a movie script, and a question. Answer the question as concisely as you can, using a single phrase if possible. Do not provide any explanation.

Story: {context}

Now, answer the question based on the story as concisely as you can, using a single phrase if possible. Do not provide any explanation.

Question: {input}

**Ours**: Blue Lou Boyle.

**Ground truth**: A mobster named "Blue Lou Boyle"

**Prompt length**: 29633

Figure 5: Model response in the second case from the LongBench dataset, illustrating multi-hop reasoning across non-contiguous document segments.

756
757
758
759
760
761
762
763
764
765
766
767
768
769
770
771
772
773
774
775
776
777
778
779
780
781
782
783
784
785
786
787
788
789
790
791
792
793
794
795
796
797
798
799
800
801
802
803
804
805
806
807
808
809

---

**Example 3**

**RepoBench-P:** Please complete the code given below.
{Context: here are some examples for the code completion task.}
{Input: the user provides the following code to be completed}
Next line of code:
**Ours:**   public static Frameset frameset(Attribute... attributes)
{
  return new Frameset(attributes);
  }
  public static Head head(Attribute... attributes) {
   return new Head(attributes);
  }
**Ground truth**:   public static Frameset frameset(Attribute...
attributes) {
**Prompt length**: 8158

---

Figure 6: Model response in the third case from the LongBench dataset, demonstrating adaptation to reverse-ordered interleaving and maintenance of narrative coherence.

> **Example 4**
>
> **SAMSum**: Summarize the dialogue into a few short sentences. The following are some examples.
> {Context: examples of dialogue summarization are provided here.}
> Dialogue: Ana: You sleeping?
> Catherine: Not yet.
> Ana: Wanna go visit grandma tomorrow? I miss her.
> Catherine: Yeah that would be nice :) I'll call you when I wake up
> Ana: Oki :) sleep well, good night.
> Catherine:  Good night, u too.
> **Ours**: Ana wants to visit her grandma tomorrow. Catherine will call her when she wakes up.
> **Ground truth**: Ana wants to visit grandma tomorrow. Catherine will go with her. She will call Anna when she wakes up.
> **Prompt length**: 9352

Figure 7: Model response in the fourth case from the LongBench dataset, highlighting robust information synthesis in challenging long-context scenarios.

