# OpenReview forum: "Long-Attention Weaving: Synthesizing Long-Context Data with High-Quality Short Data"
_ICLR.cc/2026/Conference — ICLR 2026 Conference Withdrawn Submission_

### Official Review · Reviewer_2Vd4 · 2025-10-30

**Soundness:** 2
**Presentation:** 2
**Contribution:** 2
**Rating:** 2
**Confidence:** 4

**Summary:**

This paper presents a method: long context weaving, which proposes a method to combine short documents into longer documents suitable for long-context Language Model training. The core method utilizes attention patterns of models to find documents that have longer internal dependencies. The synthesis method creates longer documents by interleaving sections from selected documents.

The authors has conducted ablation study to show the method can indeed improve the performance.

**Strengths:**

The core idea makes sense. Using attention patterns to find documents that have longer dependency, and take them apart synthetically, will intuitively enable long-context text the requires authentic and complex interaction. This method may has advantage over some synthetic methods where the dependency are unnatural.

The general idea is presented relatively clear.

The authors have put thoughts into the design of the algorithm, such as the multi-scale ranking aggregation which considers the attention scale, and the reverse-weaving method.

**Weaknesses:**

The technical details are not presented comprehensively.
  - The overall algorithm should be placed in the main document instead of the appendix.
  - The key distance threshold set K should be defined clearly. Is it the set of integers of 1: |s| (where |s| is number of tokens in a document) ? Or  they are selected thresholds based on some algorithm?
  - For equation 1 (LDS), what should be the choices for the hyperparameter $\alpha$? And the paper should also discuss why penalizes "non-uniform dependency" is needed.

There are some more methodology details of the algorithms that should be studied.
  - One specific phenomenon pointed out by the authors is the increase of perplexity on shorter context.  While the authors argue that LAW successfully compel the model to resolve dependencies beyond short windows. This degradation should have been studied more carefully. On one hand, the performance of short context is actually important and this may be an important consideration for adopting this method. On the other hand, this may also reflect that the synthetic data is not natural.

The presentation of the qualitative study needs improvement
   - In general, this section is mainly about a few attention visualization and takes significant space. This section should provide more detailed and practical insights
   - The figures are a bit hard to follow, probably due the the choice of visualization. But there are some small important details needed, for example, the axis definition should be defined early and labeled on the figures.
   - Some of the observation could have been quantitatively presented. For example, if the "band-diagonal structure" at line 402 and "visibly diminished lower-left quadrant" at line 425 are visible, I believe we can compute some metrics to provide a quantitative representation.


Another important weakness is the lack of Baselines
  - The main result table only contains comparison between the method against ablated methods, or random baselines. These baselines are too simple. Several other methods are not compared, such as LongPack [1] and ProLong [2].

[1] https://openreview.net/forum?id=tePFpDgyqg
[2] https://arxiv.org/abs/2410.02660

The effectiveness of the method:
  - Maybe it is more of a problem of perspective, but it is unclear whether LAW's 31.26 average score vs. the base model's 30.50, or vs. the random Concatenation's 31.05, is a "significant improvement". If the authors believe these gains are significant, I would suggest the improvement be put on clear perspectives.
  - As mentioned above, there are some potential concern over the short context perplexity, a study of whether short-context task performance should be evaluated. In fact, this study should be done regardless, to demonstrate that the generic model performance is not impacted.

Incomplete related work study:
  - The related work study is not comprehensive, especially for the data synthesis section. Since synthesis is the key idea of this method, this part should have been done more thoroughly. This will also help identifying relevant related work. For example, LongPack is a relevant method that synthesize longer documents, which is a very relevant comparison target.

**Questions:**

The method split the document into two halves before interleaving. I wonder if multiple sections make sense?

Would random interleave makes sense here? Consider that the reverse-order "destroy" the partial order of one document anyways, maybe random interleave will also work? Random order also may have less structural biases compared to complete reverse orders.

---

### Official Review · Reviewer_a7A8 · 2025-10-30

**Soundness:** 3
**Presentation:** 2
**Contribution:** 2
**Rating:** 4
**Confidence:** 3

**Summary:**

This paper introduces Long-Attention Weaving (LAW), a novel framework designed to address the scarcity of high-quality long-context training data for Large Language Models (LLMs). LAW synthesizes effective long-context training samples by leveraging a model's own self-attention mechanism. Extensive experiments show that continually pre-training LLaMA-2 7B on LAW-synthesized data extends its effective context window to 64k tokens and significantly outperforms strong baselines on the LongBench benchmark. The work demonstrates that attention-guided data engineering is crucial for unlocking robust long-context capabilities in LLMs.

**Strengths:**

The paper introduces a novel attention-guided approach to long-context data synthesis, which is conceptually distinct from existing heuristic or random concatenation strategies.

**Weaknesses:**

1. The experiments are limited to a single base model (LLaMA-2 7B). It would strengthen the claims to demonstrate that LAW generalizes across different model families (e.g., Qwen) or larger model scales (14B/70B).

2. The citation at line 53 appears to be incorrect: *Prolong: A data-driven approach for measuring and enhancing LLMs’ long-context proficiency. arXiv preprint arXiv:2405.15095, 2024.*

3. From Table 2, the performance gain of LAW seems minimal.

4. The evaluation benchmarks are too limited, and **LongBench** is somewhat outdated. How does LAW perform on more recent benchmarks such as **RULER**, **HELMET**, and **LongBench V2**? Moreover, prior work [1] has pointed out that using **perplexity (PPL)** as a measure of long-context performance is not reliable.


[1] Fang L, Wang Y, Liu Z, et al. What is Wrong with Perplexity for Long-context Language Modeling?[J]. arXiv preprint arXiv:2410.23771, 2024.

**Questions:**

See Weaknesses.

---

### Official Review · Reviewer_1TLp · 2025-10-30

**Soundness:** 2
**Presentation:** 3
**Contribution:** 2
**Rating:** 2
**Confidence:** 3

**Summary:**

This paper addresses the critical challenge of data scarcity for training long-context Large Language Models. The authors introduce Long-Attention Weaving (LAW), a novel, two-stage framework for synthesizing a high-quality, long-context training curriculum from a pool of shorter documents. The first stage employs a multi-scale, attention-based scoring metric to identify and select short documents that are rich in long-range dependencies, leveraging the model's own internal states as a quality signal. The second stage uses a novel "interleaving" strategy, which includes a reverse-ordering component, to weave these selected documents into complex sequences designed to force the model to learn non-trivial, long-distance relationships. The authors demonstrate through experiments on a LLaMA-2 7B model that their method extends the effective context window to 64k and outperforms baselines on the LongBench benchmark.

**Strengths:**

1. The paper presents a interesting perspective on long-context data synthesis, shifting the focus from architectural changes to a model-centric, data engineering approach.

2. The proposed framework is well-reasoned, breaking down the complex problem into two distinct and logical stages: data selection (based on attention scores) and data synthesis (via a challenging interleaving strategy).

3. The paper is clearly written, with a logical flow and helpful visualizations that effectively communicate the proposed method and its underlying mechanisms.

**Weaknesses:**

1. The primary weakness of this work is the underwhelming empirical results, which fail to convincingly demonstrate the superiority of the proposed LAW framework. For a data generation strategy, strong empirical validation is paramount. On the main LongBench benchmark, LAW achieves an average score of 31.26, a negligible improvement of only 0.21 points over the 'Random Concatenation' baseline (31.05). This suggests that the entire complex machinery of multi-scale attention filtering and reverse-ordered weaving provides almost no tangible benefit over the most naive baseline.

2. The fundamental premise of using a model's self-attention scores as a reliable proxy for inherent data quality is questionable. Self-attention mechanisms are known to exhibit idiosyncratic behaviors and biases, such as the 'attention sink' phenomenon, where high attention is paid to initial tokens regardless of content. Therefore, the scoring mechanism may be selecting documents that simply conform to the model's existing biases rather than those with genuine, challenging long-range dependencies, undermining the validity of the entire filtering stage.

**Questions:**

1. Given the very small performance improvement over the Random Concatenation baseline on LongBench, how do you justify the significant added complexity of the LAW framework?

---

### Official Review · Reviewer_4n56 · 2025-10-31

**Soundness:** 2
**Presentation:** 3
**Contribution:** 2
**Rating:** 2
**Confidence:** 3

**Summary:**

This paper proposes Long-Attention Weaving (LAW), a data-centric framework for improving long-context learning in large language models (LLMs).

The key idea is to synthesize high-quality long-context training data from existing short documents by leveraging the model’s own self-attention patterns to guide both document selection and concatenation.

LAW is evaluated by continued pretraining of LLaMA-2 7B on 1B tokens synthesized through this method, extending its effective context window from 4k to 64k.

**Strengths:**

1. The paper is written clearly, with a logical structure and well-defined methodological pipeline (dependency scoring -> ranking aggregation -> interleaved synthesis).

2. The idea of leveraging a model’s own attention patterns to guide data synthesis is conceptually appealing and aligns with recent trends in self-evaluative or model-in-the-loop data generation.

3. LAW improves perplexity at 64k tokens and outperforms naive concatenation baselines on LongBench, showing tangible gains in extended-context performance.

**Weaknesses:**

1. The proposed framework primarily rearranges short documents through attention-guided filtering and interleaving, but this process does not guarantee genuine semantic or causal dependencies across document boundaries.
As a result, the synthesized long sequences may only create token-level distance rather than meaningful long-range reasoning structure.

2. The Long-range Dependency Score (LDS) is computed from raw attention magnitudes, which may reflect stylistic or frequency biases rather than true dependency strength.
The subsequent multi-scale ranking aggregation and interleaving strategies are largely heuristic, lacking justification or formal analysis of stability.

3. Trade-off in modeling quality: The method improves perplexity at long context lengths but significantly degrades short-context performance (2k–8k).

4. Comparisons are limited to Random Concatenation and Ordered Interleaving; no evaluation is provided against stronger long-context methods. No results on Longbench v2, RULER.

**Questions:**

1. Have you tried validating LDS against any human-annotated or linguistic metrics?

2. In the interleaving stage, are documents mixed randomly, or do you use any strategy to decide which halves are combined?
If purely random, how do you ensure that the resulting “long-context” data contains meaningful cross-segment relationships?

3. The paper reports lower perplexity at 64k but higher at short context lengths.
Did you analyze whether this trade-off harms general downstream tasks that rely on shorter contexts?

4. How sensitive is LAW to the choice of distance thresholds k in LDS computation?
Would different scales or weighting schemes change the ranking outcome significantly?

5. The choice of LLaMA-2 7B as the main experimental backbone limits the paper’s contemporaneity.

---

### Note · Authors · 2025-12-01

**Comment:**

I have read and agree with the venue's withdrawal policy on behalf of myself and my co-authors.

**Withdrawal Confirmation:**

I have read and agree with the venue's withdrawal policy on behalf of myself and my co-authors.